# Identification of a novel conserved signaling motif in CD200 receptor required for its inhibitory function

**Laura M. Timmerman**[1,2], **J. Fréderique de Graaf**[1], **Nikolaos Satravelas**[3], **Çan Kesmir**[3], **Linde Meyaard**[1,2☯], **Michiel van der Vlist**[1,2☯]*

**1** Department of Immunology, Center for Translational Immunology, University Medical Center Utrecht, Utrecht University, Utrecht, The Netherlands, **2** Oncode Institute, Utrecht, The Netherlands, **3** Theoretical Biology & Bioinformatics, Science Faculty, Utrecht University, Utrecht, The Netherlands

☯ These authors contributed equally to this work.
* mvlist2@umcutrecht.nl

**Data Availability Statement:** All relevant data are within the manuscript and its Supporting Information files.

## Abstract

The inhibitory signaling of CD200 receptor 1 (CD200R) has been attributed to its NPxY signaling motif. However, NPxY-motifs are present in multiple protein families and are mostly known to mediate protein trafficking between subcellular locations rather than signaling. Therefore, we investigated whether additional motifs specify the inhibitory function of CD200R. We performed phylogenetic analysis of the intracellular domain of CD200R in mammals, birds, bony fish, amphibians and reptiles. Indeed, the tyrosine of the NPxY-motif is fully conserved across species, in line with its central role in CD200R signaling. In contrast, P295 of the NPxY-motif is not conserved. Instead, a conserved stretch of negatively charged amino acids, EEDE279, and two conserved residues P285 and K292 in the flanking region prior to the NPxY-motif are required for CD200R mediated inhibition of p-Erk, p-Akt308, p-Akt473, p-rpS6 and LPS-induced IL-8 secretion. Altogether, we show that instead of the more common NPxY-motif, CD200R signaling can be assigned to a unique signaling motif in mammals defined by: EEDExxPYxxYxxKxNxxY.

## Introduction

Inhibitory immune receptors control the magnitude and duration of an immune response and thereby prevent collateral damage [1, 2]. CD200 Receptor 1 (CD200R) is an immune inhibitory receptor that is expressed on myeloid-, T- and B-cells [3, 4]. The major ligand for CD200R is CD200 [5]. CD200 is expressed on immune cells and also non-immune cells such as vasculature and neurons [6]. Additionally, in mice two other ligands for CD200R, iSec1 and iSec2, are reported to be expressed in secretory cells of the gut [7].

CD200R signaling is shown to suppress anti-tumor immunity [8–16], and CD200R expression and function is altered in autoimmunity [17]. Besides autoimmunity and cancer, CD200R is implicated in both viral and bacterial immunity. In mouse hepatitis coronavirus, CD200R inhibits TLR7 signaling, dampening type I IFN production in response to TLR7 ligands [18], and in mouse influenza CD200R decreases inflammation during pulmonary infection [19].

**Funding:** JFG, NS and CK received no specific funding for this work. LM, LMT and MV received funding from Oncode Institute (https://www.oncode.nl/). LM received funding from the Netherlands Organization for Scientific Research (NWO; https://www.nwo.nl/; ALW Grant 821.02.025 and NWO Vici 918.15.608). MV received funding from the Netherlands Organization for Scientific Research (NWO; https://www.nwo.nl/; ALW Grant 863.14.016). The funders had no role in study design, data collection and analysis, decision to publish, or preparation of the manuscript.

**Competing interests:** The authors have declared that no competing interests exist.

Moreover, CD200R limits colonisation and proliferation of the bacterium *Francisella tularensis* by mediating the production of reactive oxygen species in neutrophils [20]. Furthermore, CD200R expression on particular dendritic cell subsets was recently found to be decreased in patients with severe COVID-19 [21].

The extracellular tail of CD200R is well described with a detailed crystal structure, revealing the interaction interface of CD200:CD200R [22] and the need for N-glycosylation for CD200-CD200R interaction [23]. In contrast, there is limited knowledge on the intracellular tail of CD200R. Unlike many known inhibitory immune receptors, CD200R does not signal through an immune tyrosine inhibitory motif (ITIM) and does not recruit canonical inhibitory phosphatases such as SHP-1 and SHP-2 [24]. Instead, CD200R contains three conserved Tyr residues at positions 286 (Y286), 289 (Y289) and 297 (Y297). Although Y286 and Y289 are conserved, their role in CD200R signaling is not entirely clear. In contrast, Y297 is essential for CD200R signaling [25, 26]. Upon phosphorylation of Y297, CD200R recruits Dok2, which subsequently recruits RasGAP [25, 27]. We recently found that CD200R inhibits both the PI3K/Akt and the MAPK/Erk pathway, where Akt-signaling is inhibited through Dok2, and Ras-signaling through the Dok2-RasGAP signalosome (Fig 1) [17]. Inhibition of Ras-signaling leads to inhibition of Erk phosphorylation at residues threonine 202/tyrosine 204 (p-Erk), followed by inhibition of IL-8 secretion [17]. At the other signaling arm, CD200R inhibits Akt, which can be phosphorylated at threonine 308 (p-Akt308) and at serine 473 (p-Akt473). p-Akt308 is a target of PI3K/PDK, while p-Akt473 is a target of MTORC2 [28]. Phosphorylation of ribosomal protein S6 (rpS6) at serine 235/236 (p-rpS6) is a downstream effect of both Erk and Akt signaling, but might also be inhibited by CD200R via a yet unknown signaling pathway [17, 29].

The essential Y297 is embedded in a so-called NPxY-motif in mammals. In existing literature, the inhibitory capacity of CD200R has been primarily attributed to this motif. However, the NPxY-motif occurs in many other proteins where it is not associated with inhibitory signaling. In contrast, at the initial identification of the NPxY-motif in 1990 [30] it was described as a docking site for proteins that contain a phosphotyrosine binding (PTB) domain. In the epidermal growth factor receptor the NPxY-motif results in downstream activating signaling [31]. Furthermore, the NPxY-motif was found in other proteins, like the low-density lipoprotein (LDL) receptor and integrins, where the NPxY-motif is required for proper receptor internalization [30, 32, 33]. Together, these data question the validity of the NPxY-motif as the defining motif responsible for the inhibitory function of CD200R. Here, we hypothesize that additional motifs or residues in the intracellular tail of the CD200R are required to exert the full inhibitory function of the CD200R. Therefore, we assessed conservation of the CD200R signaling domain by sequence analysis in five animal classes and functionally determined the contribution of conserved residues to CD200R signaling.

## Materials and methods

### Phylogenetic analysis

The sequences used for the phylogenetic analysis and sequence logos were downloaded in June-July 2020 from NCBI database using NCBIs protein BLAST tool with human CD200R1 as query. For the phylogenetic analysis all sequences with an annotation "partial" were "low quality" discarded. Moreover, for every species we took only one isoform of CD200R sequences. This process resulted in a total of 179 CD200R sequences, divided over all five animal classes. Full sequences were aligned using MUSCLE software [37]. As there is an abundance of sequences from mammals, birds and fish, we manually reduced the number of sequences from these species to 87 sequences to generate an easily readable phylogenetic tree using treeDyn software [36]. Mouse CD200 sequence was used as an outgroup to root the tree.

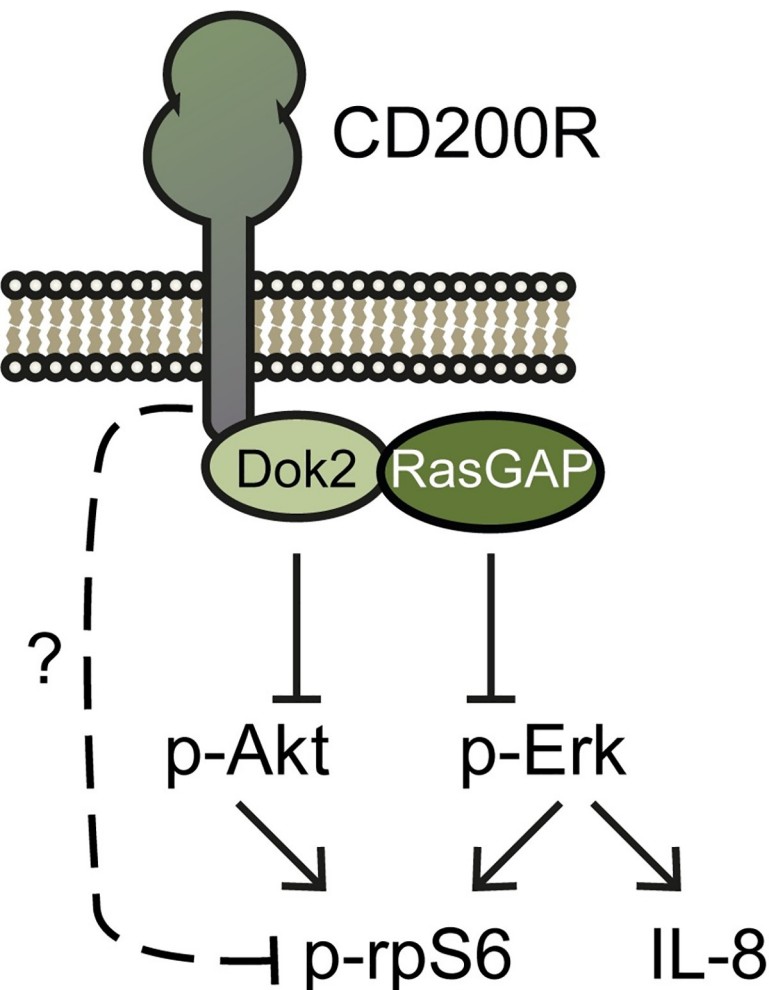

**Fig 1. Overview of the CD200R signaling pathway.** p-Akt is inhibited by Dok2, which also reqruits RasGAP. RasGAP inhibits p-Erk, resulting in inhibition of IL-8. Inhibition of p-rpS6 by CD200R is a result of the inhibition of both p-Akt and p-Erk, although CD200R-mediated p-rpS6 inhibition might also be the result of another yet unknown pathway. This figure was adapted from Van der Vlist et al. [17].

The whole set of CD200R1 sequences (n = 179) was used to create conservation plots using WebLogo [35]. Gene IDs of the sequences used to construct the tree and the conservation plots can be found in S1 File. For specific numbers of sequences per animal class used for the phylogenetic tree and the conservation plots, see S1 Table.

*In silico* predictions of possibly damaging amino acid substitution in the intracellular tail of human CD200R were generated using PROVEAN [38] and PANTHER [39].

## Cell lines

U937 (human, male origin, not authenticated) cells were maintained in RPMI (Gibco) with 10% fetal calf serum (FCS), 1% Penicillin-Streptomycin (P/S) and 200 mM Glutamine in a 5% $CO_2$ humidified 37°C incubator. Before experiments, U937 cells were differentiated with 30 ng/ml Phorbol 12-myristate 13-acetate (PMA; Sigma) in RPMI with 5% FCS, 1% P/S and 200 mM Glutamine for 24 hours, followed by a 24-hour rest period in RPMI with 5% FCS after washing.

**Table 1. Schematic representation of primers used for mutagenesis.**

|  | Outer Forward | Inner Forward | Inner Reverse | Outer Reverse | PCR type |
|---|---|---|---|---|---|
| Truncated | CD200R_F | - | - | CD200R_tr | Sewing |
| Y0 | - | Y0_F | Y0_R | - | Quick change |
| EEDE279 | CD200R_F | EEDE_F | EEDE_R | CD200R_R | Sewing |
| P285 | CD200R_F | P285_F | P285_R | CD200R_R | Sewing |
| K292 | - | K292_F | K292_R | - | Quick change |

Primer sequences are depicted in Table 2.

## U937-CD200R mutagenesis and retroviral transduction

Wild type CD200R1 and CD200R1 mutants Y286F, Y289F and Y297F were generously provided by Marion Brown in pHR-SIN-BX-IRES-Em [25]. To generate Y267F and K292A, we Quick Changed CD200R in pHR-SIN-BX-IRES-Em with 5 ng/µl DNA template, 0.5 µM forward and reverse primer, 200 µM dNTPs, 0.2 µl F-320S polymerase and 5x HF buffer (Thermo Scientific) in 20 µl total reaction volume. 1% DMSO was used for primers with Tm > 72 ˚C. PCR products were digested by Dnp1 (New England Biolabs). After Quick Change, we amplified all CD200R variants from pHR-SIN-BX-IRES-EM, and ligated the product in pMXneo (see below). Truncated CD200R, P285A and 279TTST mutants were made using Sewing PCR. All primers used for PCR are depicted in Tables 1 and 2.

All constructs were purified from the agarose gel (QIAEX II Gel Extraction Kit, Qiagen), and digested by NotI and Bgl-II (New England Biolabs). The digested fragments were ligated with T4 DNA ligase (Thermo Scientific) into the retroviral vector pMXneo, which was opened with NotI and BamHI (New England Biolabs). Sequences of all inserts in pMXneo were validated by Sanger sequencing.

The constructs were transfected into Phoenix retroviral packaging cells using Fugene (Promega) and the pCL-Ampho retrovirus packaging vector in Opti-MEM culture medium. U937 cells were transduced by culturing them in the cell-free supernatant of the transfected cells supplemented with polybrene. Cells were FACS-sorted on high CD200R expression, and treated with puromycin (Thermo Scientific) just after FACS-sort to prevent infections and cultured as described above.

**Table 2. Primer sequences for mutagenesis and sub-cloning.**

| Name | Sequence |
|---|---|
| CD200R_F | taagcaAGATCTGAGAAAACAGAAATGCTCTGC |
| CD200R_R | attcgtGCGGCCGCTTATAAAGTATGGAGGTCTGTGTC |
| CD200R_tr | attcgtGCGGCCGCTTAAACAACTGGAGTAGATTCTGTTTTATTC |
| Y0_F | CAATGGCTGCAGAAAATTTAAATTGAATAAAAC |
| Y0_R | GTTTTATTCAATTTAAATTTTCTGCAGCCATTG |
| EEDE_F | CTCCAGTTGTTACGACGTCTACAATGCAGCCC |
| EEDE_R | GGGCTGCATTGTAGACGTCGTAACAACTGGAG |
| P285_F | GATGAAATGCAGGCCTATGCCAGC |
| P285_R | GCTGGCATAGGCCTGCATTTCATC |
| K292_F | CCTACACAGAGGCGAACAATCCTCTC |
| K292_R | GAGAGGATTGTTCGCCTCTGTGTAGG |

The names in the first column correspond to the names in Table 1. Small letters are aspecific overhang to facilitate digestion, capital underlined are digestion sites.

## Stimulation assays with U937-CD200R mutants

PMA differentiated U937 cells were seeded in round bottom plates and incubated with anti-CD200R (in-house, clone OX108) or anti-SIRL-1 (in-house, clone 1A5) as isotype control at 3 μg/ml for 60 minutes in a 5% $CO_2$ humidified 37˚C incubator. For intra-cellular staining, cells were subsequently fixed (see "PhosFlow" for details). For IL-8 secretion, cells were stimulated with 200 ng/ml LPS (Salmonella Typhosa, Sigma) and cultured for 24 hours in a 5% $CO_2$ humidified 37˚C incubator before harvesting cell-free supernatant for ELISA.

## Intracellular flow cytometry for phosphorylated proteins (PhosFlow)

For analyses of phosphorylated proteins with flow cytometry, cells were fixed with a final concentration of 3% PFA (VWR) in medium for 10 minutes at room temperature. Cells were harvested, transferred into a 96-well V-bottom plate and washed, and subsequently permeabilized with 100% ice cold methanol, incubated for at least 5 minutes at -20˚C and stored for a maximum of 3 months at -20˚C. Methanol fixed cells were washed twice in flow cytometry buffer (PBS, 1% BSA, 0.01% sodium azide, from here on called FACS buffer). Staining was performed in FACS buffer and non-specific binding was prevented by adding 5% normal donkey serum to FACS buffer. All antibodies used are depicted in Table 3. The primary antibodies were incubated shaking for 16 hours at 4˚C. Cells were washed three times and incubated with a

**Table 3. Overview of materials.**

| Antibodies | | | | | | |
|---|---|---|---|---|---|---|
| Target | Label | Vendor | Clone | Catalog# | Usage | RRID |
| CD200R1 | Unlabeled | In house | OX108 | - | 3 ug/ml | n.a. |
| Isotype/SIRL1 | Unlabeled | In house | 1A5 | - | 3 ug/ml | n.a. |
| Akt p-Thr308 | Unlabeled | CST | D25E6 | 13038 | 3600x | AB_2629447 |
| Akt p-Ser473 | Unlabeled | CST | D9E | 4060 | 200x | AB_2315049 |
| Erk pThr402/Tyr204 | Unlabeled | CST | D13.14.4E | 4370 | 200x | AB_2315112 |
| rpS6 pSer234/235 | PE | CST | D57.2.2E | 5316 | 200x | AB_10694989 |
| CD200R1 | Unlabeled | R&D | Poly clonal | AF3414 | 200x | AB_2228955 |
| Donkey-a-Rabbit | AF594 | ThermoFisher | Poly clonal | A21207 | 400x | AB_141637 |
| Donkey-a-Goat | AF647 | Jackson | Poly clonal | 705-607-003 | 200x | AB_2340439 |
| Donkey-a-Goat | AF488 | Life technologies | Poly clonal | A11055 | 400x | AB_2534102 |
| CD200R | PE | eBioscience | OX108 | 12 9201 | 200x | AB_1210851 |

| Experimental Models: Cell Lines | Source |
|---|---|
| Human: U937 | Kind gift from Dr. J. Leusen (UMC Utrecht) |

| Devices and software | Source |
|---|---|
| BD Fortessa (4-laser) flow cytometer | BD Bioscience |
| Chemidoc XRS+ | Biorad |
| Clariostar plate reader | BMG LABTECH |
| BD FACSDiva | BD Bioscience |
| FlowJo | BD Bioscience |
| Prism 8.3 | GraphPad |
| ProteinBLAST | https://blast.ncbi.nlm.nih.gov/Blast.cgi?PAGE=Proteins [34] |
| WebLogo | http://weblogo.threeplusone.com [35] |
| treeDyn | http://www.treedyn.org [36] |
| MUSCLE | https://www.ebi.ac.uk/Tools/msa/muscle [37] |
| PROVEAN Protein (v.1.1.3) | http://provean.jcvi.org/seq_submit.php [38] |
| PANTHER (version 9.0) | http://www.pantherdb.org/tools/csnpScore.do [39] |

conjugated secondary antibody for 1 hour at room temperature while shaking, washed three times, and fluorescence was assessed on a BD Fortessa using a high throughput sampler attached to the flow cytometer.

## ELISA

IL-8 secretion was measured using an ELISA kit (Life Technologies; 88-8086-88). Cell-free supernatant was harvested and stored at -20°C until assayed by ELISA. The manufacturer's protocol was followed with few exceptions. Half of the recommended volumes were used throughout the protocol, except for wash buffer and blocking buffer. The standard curve and assay samples were incubated for 2 hours at room temperature while shaking. Optical densities (OD) were measured using a Clariostar plate reader. Using PRISM 8.3 we used a 4-parameter dose response curve of the standard curve to extrapolate the unknown concentrations.

## Calculations

The percentage of inhibition by CD200R was calculated as follows:

$$(1 - (anti-CD200R/average\ isotype\ control)) * 100\%$$

## Statistics

In all figure legends we have indicated the statistical test used to determine significance, and the n of experiments. Statistical analysis was performed in Prism 8.3.

## Results

### Conservation of the intracellular CD200R domain is not limited to the NPxY-motif

We used human CD200R1 as input in the NCBI protein BLAST and identified 179 homologues of various species (S1 File). These species included mammals, birds, bony fish, reptiles and amphibians. Phylogenetic analysis revealed that CD200R clustered according to the expected evolutionary patterns of species (Fig 2A). To determine conservation of the intracellular CD200R signaling domain, we focused on the region enclosing the NPxY-motif. As expected, we found the three conserved Tyr residues Y286, Y289 and Y297, and the Asn of the NPxY-motif (N294) (Fig 2B). In contrast, the Pro of the NPxY-motif (P295) was not found conserved amongst all species analyzed (Fig 2B). Instead, we found that several surrounding residues were highly conserved, including a stretch of the negatively charged amino acids Glu and Asp between positions 278 and 282 (E/D278-282), the Pro at position 285 (P285), the Ser/Thr at position 288 (S/T288) (both polar amino acids that can be phosphorylated, depicted in green in the sequence logos of Fig 1 to express this similarity), the Lys/Arg at position 292 (K/R292) (both positively charged amino acids, depicted in blue in Fig 1B–1E) and the Ile/Leu at position 296 (I/L296) (both hydrophobic amino acids, depicted in black in Fig 1B–1E). To examine whether there are differences in conservation between animal classes, we also plotted conservation for mammals, birds and bony fish only (Fig 2C–2E). For reptiles and amphibians the number of species with a CD200R sequence identified was too low for separate reliable analysis of conservation. In mammals, the stretch of negatively charged amino acids starts at position 279, compared to position 278 in the overall logo, and is here represented as a conserved EEDE-motif (EEDE279) (Fig 2C). The S/T288 is highly conserved within particular animal classes: in mammals and bony fish a conserved Ser was found, whereas in birds a conserved Thr was found at that position (Fig 2C–2E). High conservation within particular animal classes when conservation is shared between two amino acids in the overall logo also

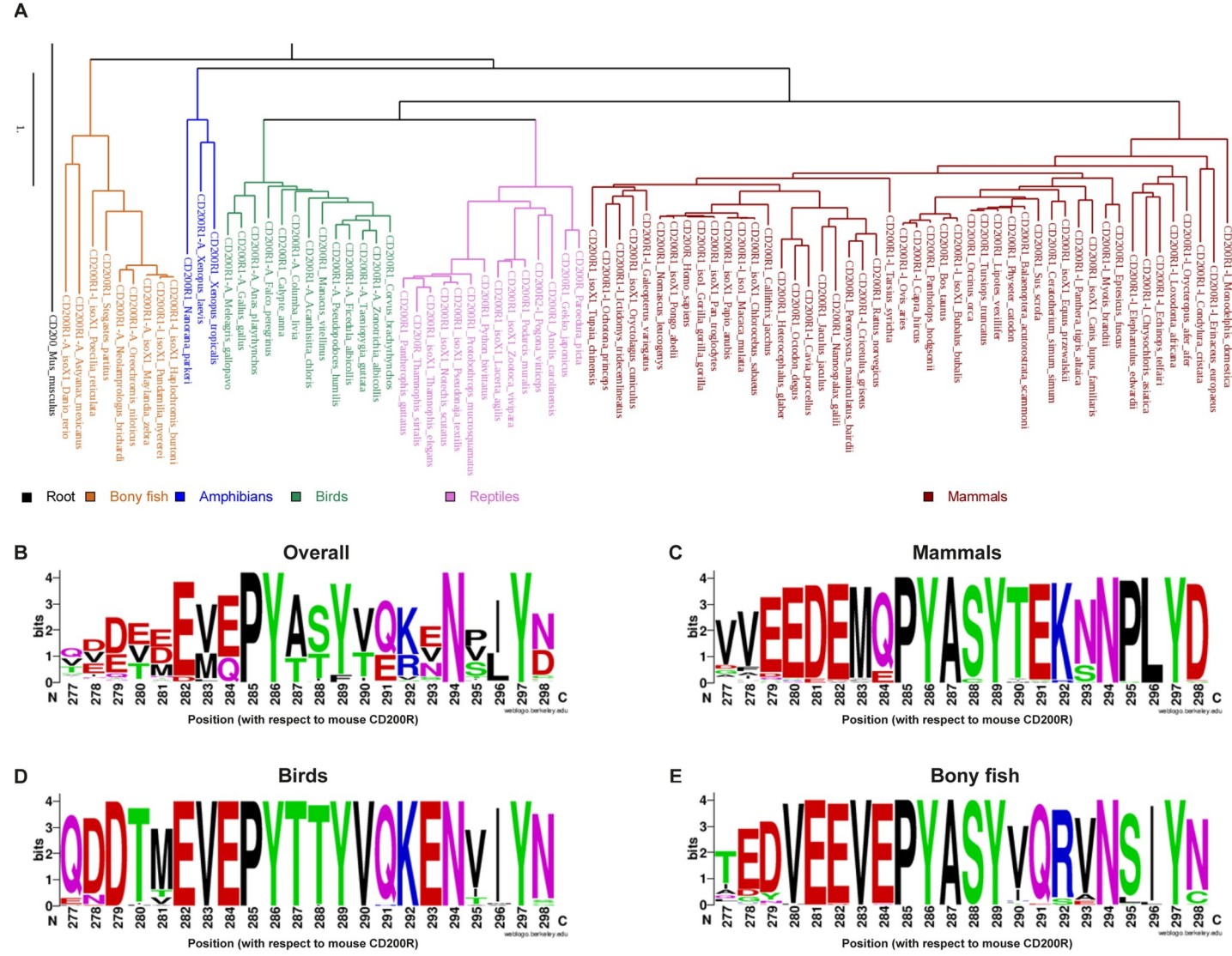

**Fig 2. The high conservation of the intracellular domain of CD200R among species is not limited to the NPxY-motif.** The NCBI protein BLAST tool identified 179 human CD200R1 homologues amongst all 5 animal classes. **A)** Phylogenetic analysis of a balanced selection of full CD200R1 homologues. Mouse CD200 sequence was used as an outgroup to root the tree. **B)** All CD200R sequences identified in A) were plotted in a conservation plot using WebLogo [35]. **C-E)** Conservation logo's created by WebLogo [35] per animal class: mammals (C); birds (D); and bony fish (E). Maximal conservation represents 4.2 bits. The height of the amino acid letters shows the frequency of the amino acid on that particular position in the sequence alignment. The amino acids are colored according to the physio-chemical properties of their side chains: polar (green), neutral (purple), basic (blue), acidic (red), or hydrophobic (black).

holds true for K/R292 and I/L296 (Fig 2B–2E). In mammals and birds a conserved Lys was found at position 292 (K292), whereas in bony fish a conserved Arg was found. In mammals position 296 contains a Leu, while bird and bony fish CD200R have a Ile on that particular position. A notable difference between animal classes is that P295 of the NPxY-motif is only conserved in mammals, but is variable in bird, bony fish, reptile and amphibian sequences (Fig 2B–2E and S1 and S2 Files).

Additionally, PROVEAN [38] *in silico* analysis considered the substitutions N294A and P300A deleterious, while PANTHER [39] *in silico* analysis predicted the substitutions P285A, Y286F, Y289F and Y297F to be potentially damaging (S3 File).

Overall, our sequence analysis shows that parts of the intracellular domain of CD200R are highly conserved throughout all species analyzed, and suggests a functional role for these residues.

## Expression of mutated forms of CD200R

We mutated a selection of highly conserved residues to investigate their contribution to CD200R function (Fig 3A). Where possible, structurally similar substitute amino acids were used: i.e. E and D were replaced by T and S respectively. The Tyr at position 267 (Y267) was included because it is conserved in all primates and bats shown in the phylogenetic tree, even though it is not present in other mammals or other animal classes (Fig 3A and S1 File). We overexpressed mutant CD200R in monocytic U937 cells that support CD200R function [17, 25, 40] to test the contribution of the different residues to CD200R function. After transduction, all CD200R mutants were expressed at the cell surface (Fig 3B), albeit in slightly different amounts.

## Fully effective CD200R signaling requires all three highly conserved tyrosine residues

CD200R ligation leads to downstream inhibition of both the PI3K/Akt and the MAPK/Erk pathway (Fig 1). We therefore ligated WT CD200R and the CD200R mutants with an agonistic

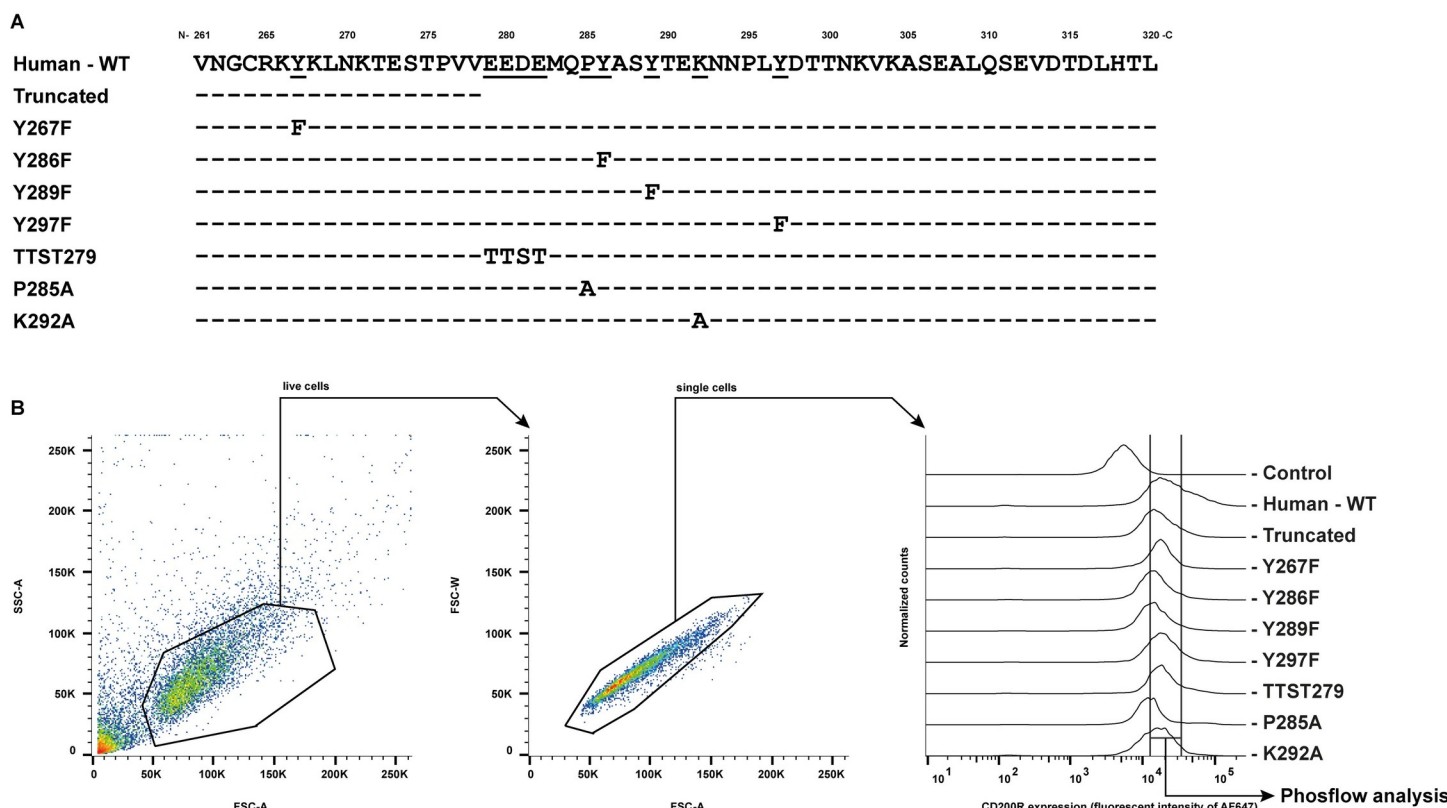

**Fig 3. U937-CD200R mutants express CD200R on their surface. A)** Conserved residues of CD200R in mammals were mutated as indicated. **B)** U937 cells were transduced with different CD200R mutants and tested for their CD200R surface expression by flow cytometry. Representative figures of the gating strategy are shown. First the population of live cells was selected. From these live cells, single cells were selected and from these single cells CD200R expressing cells were selected for phosflow analysis. SSC-A: side scatter area; FSC-A: forward scatter area; FSC-W: forward scatter width. Control: staining with the secondary antibody only.

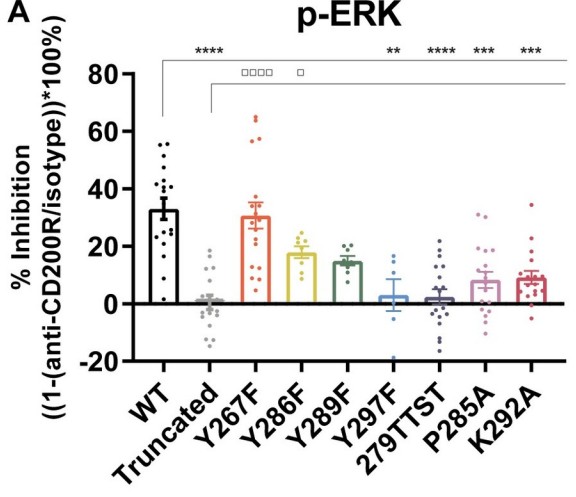

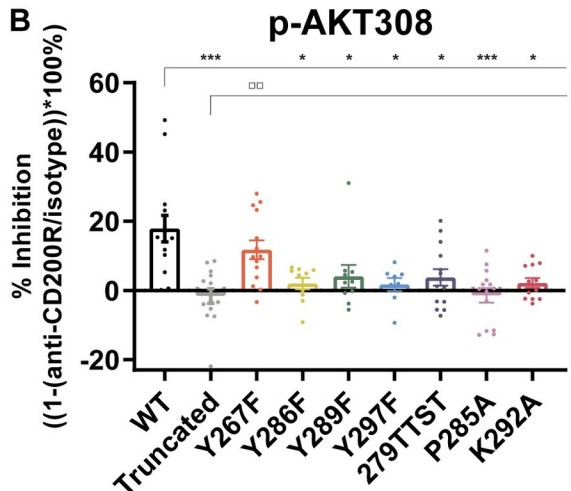

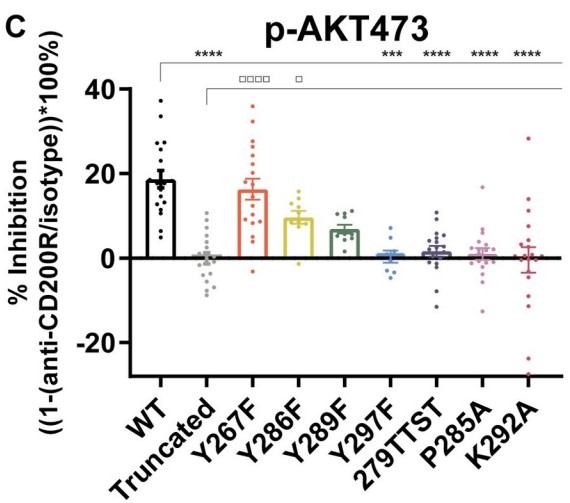

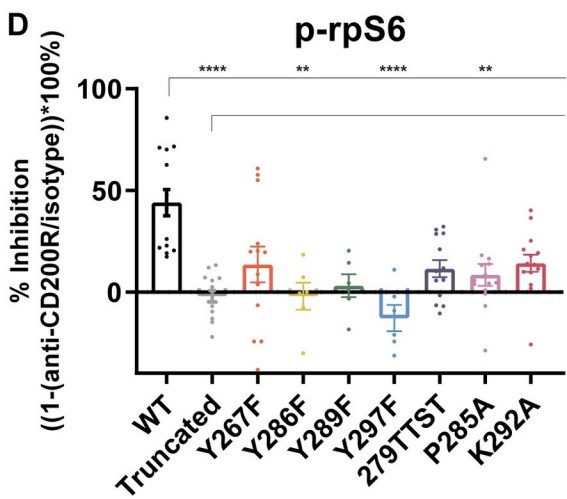

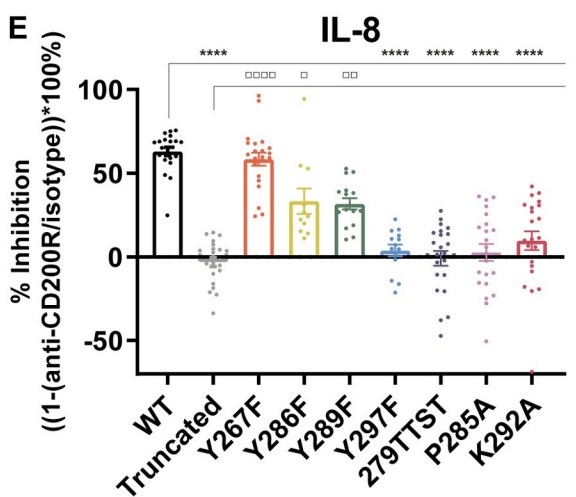

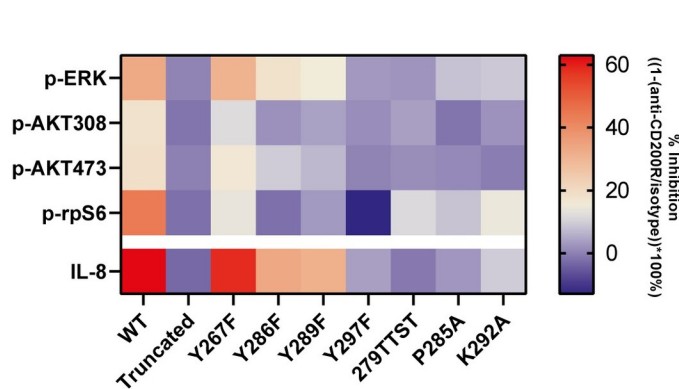

**Fig 4. EEDE279, P285 and K292 are important for CD200R function. A-E)** U937-CD200R mutants were stimulated with an agonistic CD200R antibody (OX108). p-Erk (B), p-Akt308 (C), p-Akt473 (D) and p-rpS6 (E) of CD200R$^+$ cells were assessed by phosflow analysis. IL-8 secretion after 24 hours of LPS stimulation was measured by ELISA (F). The percentage of inhibition is depicted as the relative difference between isotype and anti-CD200R stimulated U937 cells. Error bars represent the mean with SEM. All mutants were tested for significant differences compared to WT (*) and Truncated (□) using a Kruskal-Wallis test. N≥6. For the specific amount of experiments conducted per mutant and per residue, see S2 Table. **F)** Heatmap of the means of the data shown in Fig B-F. A double gradient from blue to red was used for the percentage of inhibition.

antibody (clone OX108) in U937 cells, which have constitutive phosphorylated Erk, Akt and rpS6. With intracellular flow cytometry we quantified p-Erk, p-Akt308, p-Akt473 and p-rpS6. We only analyzed cells with similar CD200R expression by applying an CD200R-expression gate in our phosflow analysis. This gate was set as indicated in Fig 3B.

Wild type (WT) CD200R inhibited p-Erk, p-Akt308, p-Akt473 and p-rpS6 (Fig 4A–4D). Truncation of the intracellular tail of CD200R abrogated inhibition of p-Erk, p-Akt308, p-Akt473 and p-rpS6, demonstrating that our assay is specific for CD200R signaling. In line with the recruitment of Dok2 to Y297, Y297F completely failed to inhibit p-Erk, p-Akt308, p-Akt473 and p-rpS6 (Fig 4A–4D). Y286F and Y289F did have little or no effect on the ability of CD200R to inhibit p-Erk and p-Akt473 (Fig 4A and 4C). On the other hand, these mutations were not able to inhibit p-Akt308 and p-rpS6, suggesting that Y286 and Y289 are only required for inhibition of the PI3K/Akt pathway (Fig 4B and 4D). Y267, the most membrane-proximal tyrosine, is conserved in primates and bats but not in other species suggesting it is redundant for CD200R function. Indeed, Y267F did not affect the ability of CD200R to inhibit p-Erk, p-Akt308, p-Akt473 and p-rpS6. Together, these data show that Y286, Y289 and Y297 are not only conserved, but also required for fully effective CD200R signaling.

## EEDE279, P285, and K292 play an important role in CD200R signaling

Besides the conserved Tyr residues, we identified the highly conserved E/D278-282 motif and the highly conserved residues P285 and K/R292 in the intracellular domain of CD200R. We found that 279TTST, P285A, and K292A failed to efficiently inhibit p-Erk, p-Akt308, and p-Akt473 (Fig 4A–4D).

The requirement of 279TTST, P285A, and K292A for phosphorylation of rpS6 is less straight-forward: 279TTST and K292A did neither significantly differ from WT nor truncated CD200R, suggesting an intermediate phenotype. P285A, on the other hand, did result in a reduced capacity of CD200R to inhibit p-rpS6. Altogether, we conclude that not only the Tyr residues, but also EEDE279, P285 and K292 are important for CD200R signaling.

## EEDE279, P285, and K292 are important for the inhibitory function of CD200R

We also assessed the effect of the mutations on the ability of CD200R to inhibit LPS-induced IL-8 secretion by ELISA as a functional readout for inhibition of immune responses. As expected, WT CD200R inhibited LPS-induced IL-8 secretion, while truncation of the intracellular tail of CD200R abrogated this inhibition (Fig 4E). Corresponding with its effects on signaling, Y297F failed to inhibit LPS-induced secretion of IL-8. In line with literature, we found Y286F and Y289F to be about 50% less efficient in CD200R mediated inhibition of LPS-induced IL-8 secretion [25, 41]. As shown in Fig 3B–3E, Y267F did not affect CD200R signaling, suggesting that Y267 is not required for the inhibitory function of CD200R. Indeed, mutating Y267 did not change the inhibition of LPS-induced IL-8 secretion (Fig 4E). Coherent with our data on p-Erk, p-Akt308 and p-Akt473, 279TTST, P285A and K292A all failed to inhibit LPS-induced IL-8 secretion, suggesting that EEDE279, P285, and K292 are not only required for fully effective CD200R signaling, but also play a role in CD200R function.

In conclusion, although quite some variation was observed between experiments we show that Y267 does not play a role in both CD200R signaling and function, and that Y286 and Y289 are mainly important for CD200R mediated inhibition of the PI3K/Akt pathway (Fig 4F). In contrast, EEDE279 and K292 are required for inhibition of the MAPK/Erk pathway, and to a lesser extent for the inhibition of the PI3K/Akt pathway. Furthermore, we found that Y297 and P285 are of high importance for both CD200R signaling and function, as mutating them yielded similar results as truncation of the entire CD200R signaling domain (Fig 4F).

## Discussion

Signaling through an ITIM is considered to be the canonical signaling pathway for inhibitory receptors. Most inhibitory receptors have one or more ITIMs, which recruit SH2 domain-containing phosphatases, leading to downstream inhibitory signals [42, 43]. CD200R on the contrary, is a unique inhibitory immune receptor that signals through a non-canonical inhibitory motif [25]. Here, coherent with literature [25], we found Y297 to be conserved among all mammals we analyzed. However, our phylogenetic analysis revealed that although multiple residues in the intracellular tail are highly conserved, P295 of the NPxY-motif is not. We found that besides mammals in all other animal classes P295 is replaced by another amino acid, most frequently a Val, Ser, or Thr. The introduction of a Val, Ser or Thr at this position changes the PTB binding domain into an NxxY PTB-domain [44, 45]. The NxxY-motif of the VEGFR3 does interact with Dok2, and therefore the absence of P295 may not necessarily impact the ability of CD200R to recruit Dok2 [26]. Of note, the replacement of P295 by a Val, Ser of Thr in combination with the Val at position 300 that is present specifically in birds, introduces a consensus ITIM (V/S/TxYxxV), which overlaps with the NxxY-motif. This may suggest that CD200R in birds potentially signals through an additional ITIM-pathway. However, based on full conservation of the NxxY-motif and additional membrane proximal residues (K/R292 and P285) among all analyzed species, we speculate that the membrane distal ITIM in birds is not functional.

In addition to the conservation of Y297, we recapitulated that the two other Tyr residues, Y286 and Y289, are conserved and required for CD200R signaling, further pointing to the NPxY-motif not being the single determinant of CD200R function. We found an additional highly conserved E/D278-282 motif, and additional highly conserved residues membrane-proximal of the fully conserved NxxY-motif in the cytosolic domain of CD200R. To assess their role in CD200R signaling and function we conducted phosflow analysis and measured LPS-induced IL-8 secretion. For phosflow analysis we corrected for differences in CD200R expression during the analysis of the flow cytometry data, which was not possible for IL-8 secretion. However, the effects of the mutations show a similar pattern in Erk phosphorylation and LPS-induced IL-8 secretion. CD200R-mediated inhibition of Erk was previously shown to be essential for inhibition of IL-8 [25]. Therefore, we do not think (minor) expression differences have a major impact on our conclusion. For reasons we do not understand, both phosflow analysis and the IL-8 secretion measurements showed variation between experiments. However, as the number of experiments is high, we believe that the mean values are representative for the real biological situation. Mutagenesis of EEDE279, P285, and K292 negatively affects the inhibitory capacity of CD200R in a mammalian cell line. Given that E/D278-282, P285, and K/R292 are highly conserved amongst all species analyzed, we assume that their function in other animals than mammals is comparable to the function of EEDE279, P285, and K292 in mammals, respectively. Without further in-depth analysis, we can only speculate how these residues are involved in CD200R signaling: they could facilitate binding of tyrosine kinases upstream of Dok2 recruitment or facilitate binding of Dok2 downstream of tyrosine phosphorylation. For example, they could be important for the structure of the intracellular

tail of CD200R to support signaling. Lysines and arginines can be involved in hydrogen bonds and/or can undergo posttranslational modifications [46, 47], whereas prolines are often part of tight turns in protein structures [46], and the negatively charged E/D278-282 could play a role in protein stability by the formation of salt-bridges [46]. Altogether, E/D278-282, P285, and K/R292 are not only highly conserved, but also relevant for the signaling pathway of CD200R and its inhibitory function.

Our analysis revealed that Y286 and Y289 of CD200R are required to inhibit p-Akt308 and p-rpS6. In contrast, as previously reported, they are of relatively low importance for inhibition of p-Erk and IL-8 secretion [25, 41]. This suggests that these residues have limited impact on RasGAP recruitment, but possibly do affect recruitment of another downstream adaptor that targets the PI3K/Akt pathway. This is in line with our previous finding that in absence of Ras-GAP, CD200R is able to inhibit the Akt pathway [17]. Multiple phosphatases can dephosphorylate and/or prevent phosphorylation of p-Akt308. PTEN (mutated in U937 [48]) and SHIP prevent activation of PDK, upstream of p-Akt308. SHIP has been shown to be recruited to CD200R in mouse mast cells [27], but this was not replicated in U937 [40]. Alternatively, PP2A-B55a directly dephosphorylates p-Akt308 [49], but recruitment/activation of any of these phosphatases for CD200R remains to be established.

Overall, our data show that the signaling motif of CD200R is highly conserved and not solely defined by the NPxY-motif. We identified additional residues that are involved in CD200R signaling, which extends the CD200R signaling motif in mammals to EEDExxPYx-xYxxKxNxxY. However, it is possible that not every 'x' of the motif can be any residue, since the conserved residues S/T288 and I/L296 were not examined for their function in CD200R signaling and function. The newly defined E/D278-282 motif might also play a role in other inhibitory receptors that lack an ITIM, such as lymphocyte activation gene-3 (LAG3) [50]. The signaling domain of LAG3 contains a highly conserved KIEELE domain, which resembles E/D278-282 found in CD200R. Moreover, the newly identified signaling motif could be of importance in integrins, which also contain an NPxY-motif [30, 32, 33]. In the intracellular tail of integrin $\alpha_{IIb}\beta_3$ a stretch of three glutamates can be found membrane proximal of the NPxY-motif at a similar distance as E/D278-282 and the NPxY-motif of the CD200R [51]. Furthermore, integrin $\alpha_{IIb}\beta_3$ is associated with Dok2, as is CD200R [52, 53]. Altogether, further studies are needed to reveal whether CD200R is unique, or that multiple receptors utilize similar motifs to regulate immune responses.

## Supporting information

**S1 File. Identified CD200R homologues.**
(XLSX)

**S2 File. P295 is only conserved in mammals.**
(XLSX)

**S3 File. *In silico* predictions of the pathogenic effect of CD200R mutations.**
(XLSX)

**S1 Table. Number of species used per animal class for the phylogenetic tree (Fig 2A) and conservation logos (Fig 2B–2E).** A single sequence per species was used.
(DOCX)

**S2 Table. Numbers of individual experiments per CD200R mutant and per residue shown in Fig 4.** TR = Truncated.
(DOCX)

**S1 Data. Raw data and calculations of the data shown in Fig 3B and Fig 4.**
(XLSX)

## Acknowledgments

We would like to thank Marion Brown (University of Oxford) for kindly providing us CD200R plasmids and the Core Flow cytometry Facility of the Center for Translational Immunology for their help with FACS sorting. We also would like to thank all members of the Meyaard lab (UMC Utrecht) for valuable discussions.

## Author Contributions

**Conceptualization:** Linde Meyaard, Michiel van der Vlist.

**Data curation:** Laura M. Timmerman.

**Formal analysis:** Laura M. Timmerman, J. Fréderique de Graaf, Nikolaos Satravelas, Çan Kesmir, Michiel van der Vlist.

**Funding acquisition:** Linde Meyaard, Michiel van der Vlist.

**Investigation:** Laura M. Timmerman, J. Fréderique de Graaf, Nikolaos Satravelas, Çan Kesmir, Michiel van der Vlist.

**Methodology:** Laura M. Timmerman, J. Fréderique de Graaf, Nikolaos Satravelas, Çan Kesmir, Michiel van der Vlist.

**Software:** Nikolaos Satravelas, Çan Kesmir.

**Supervision:** Çan Kesmir, Linde Meyaard, Michiel van der Vlist.

**Validation:** Laura M. Timmerman, J. Fréderique de Graaf, Michiel van der Vlist.

**Visualization:** Laura M. Timmerman.

**Writing – original draft:** Laura M. Timmerman.

**Writing – review & editing:** Çan Kesmir, Linde Meyaard, Michiel van der Vlist.

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
