## [Decision Letter · Decision Letter 0]

14 Jan 2021

PONE-D-20-39266

Identification of a novel conserved signaling motif in CD200 Receptor required for its inhibitory function

PLOS ONE

Dear Dr. van der Vlist,

Thank you for submitting your manuscript to PLOS ONE. After careful consideration, we feel that it has merit but does not fully meet PLOS ONE’s publication criteria as it currently stands. Therefore, we invite you to submit a revised version of the manuscript that addresses the points raised during the review process.

We look forward to receiving your revised manuscript.

Kind regards,

Myeongwoo Lee, Ph.D.

Academic Editor

PLOS ONE

Journal Requirements:

Additional Editor Comments:

Please accept my apologies for delay in review process. Attached are comments of reviewers. Please revise your manuscript according to their comments. Thank you.

Reviewers' comments:

Reviewer's Responses to Questions

**Comments to the Author**

1. Is the manuscript technically sound, and do the data support the conclusions?

Reviewer #1: Yes

Reviewer #2: Yes

2. Has the statistical analysis been performed appropriately and rigorously? 

Reviewer #1: Yes

Reviewer #2: Yes

3. Have the authors made all data underlying the findings in their manuscript fully available?

Reviewer #1: Yes

Reviewer #2: Yes

4. Is the manuscript presented in an intelligible fashion and written in standard English?

Reviewer #1: Yes

Reviewer #2: Yes

5. Review Comments to the Author

Reviewer #1: This work presents the significance of NPxY-motif in CD200R using the relatively new flow cytometry technology to make an effective record for visualization of research details. The authors demonstrate the step-by-step selecting the residue of interest in the NPxY motif, introducing the substitution, and effect of the single amino acid change in the human cell line.

1. To support Figure 1 result, I would recommend including prediction result of pathogenic effect with the substitution using public In-Silico analysis tools i.e., PANTHER, PROVEAN, PolyPhen2, SNAP2, or Mutation Assessor.

2. Line 210 mentioned that the different expression levels between each mutant. However, describing the method of how to normalize the difference is not sufficient.

3.Line 318-323. These statements need to be rewritten with scientific interpretation of data reflecting the rate of the positive result.

Reviewer #2: The authors identified novel conserved motifs on the cytoplasmic tail of CD200R. To my understanding, they ran the NCBI protein BLAST among different species. They made point mutations on these conserved motifs to test their roles on CD200R inhibitory function. This manuscript should be revised before accepted for publication.

1. This reviewer finds the abstract on the cover page differently compared to lines 14 – 28. I do not know if this is allowed in the editorial policy of PLOS ONE.

2. Introduction. I like to see more writings about the mechanism of CD200 and CD200R interaction in the immune response context. I will elaborate and place Figure 3A in the introduction section. For the general audience, the authors must provide CD200R involvement in 'inhibitory signaling.' I understood this manuscript is about the inhibition of inhibitory signaling. Readers need to know how the wild type cytoplasmic tail inhibits the downstream signaling before approaching the primary data.

3. Figure 2B needs more explanation. Figure 2B needs labels on the X- and Y-axis, the relationship among these three plots, and what those peaks between control and mutants mean.

4. Line 210. We need to explain how to make sure all mutant proteins are expressed at the cell surface from Figure 2B.

5. Line 230-232 need more explanation about how to correct for differences in CD200R expression between mutants. How does the CD200R expression gate was set for Phosflow analysis?

6. Line 274: what does ‘low importance’ mean?

6. PLOS authors have the option to publish the peer review history of their article (what does this mean?). If published, this will include your full peer review and any attached files.

Reviewer #1: No

Reviewer #2: No

---

## [Author Response · Author response to Decision Letter 0]

16 Feb 2021

We responded to the comments of the reviewer and the editor in a point-to-point response, which is added as a Word document named "Response to Reviewers"

---

## [Decision Letter · Decision Letter 1]

2 Mar 2021

Identification of a novel conserved signaling motif in CD200 Receptor required for its inhibitory function

PONE-D-20-39266R1

Dear Dr. van der Vlist,

We’re pleased to inform you that your manuscript has been judged scientifically suitable for publication and will be formally accepted for publication once it meets all outstanding technical requirements.

Kind regards,

Myeongwoo Lee, Ph.D.

Academic Editor

PLOS ONE

Additional Editor Comments (optional):

Reviewers' comments:

Reviewer's Responses to Questions

**Comments to the Author**

1. If the authors have adequately addressed your comments raised in a previous round of review and you feel that this manuscript is now acceptable for publication, you may indicate that here to bypass the “Comments to the Author” section, enter your conflict of interest statement in the “Confidential to Editor” section, and submit your "Accept" recommendation.

Reviewer #1: All comments have been addressed

Reviewer #2: All comments have been addressed

2. Is the manuscript technically sound, and do the data support the conclusions?

Reviewer #1: Yes

Reviewer #2: Yes

3. Has the statistical analysis been performed appropriately and rigorously? 

Reviewer #1: Yes

Reviewer #2: Yes

4. Have the authors made all data underlying the findings in their manuscript fully available?

Reviewer #1: Yes

Reviewer #2: Yes

5. Is the manuscript presented in an intelligible fashion and written in standard English?

Reviewer #1: Yes

Reviewer #2: Yes

6. Review Comments to the Author

Reviewer #1: (No Response)

Reviewer #2: (No Response)

7. PLOS authors have the option to publish the peer review history of their article (what does this mean?). If published, this will include your full peer review and any attached files.

Reviewer #1: No

Reviewer #2: No

---

## [Editor Report · Acceptance letter]

8 Mar 2021

PONE-D-20-39266R1 

Identification of a novel conserved signaling motif in CD200 Receptor required for its inhibitory function 

Dear Dr. van der Vlist:

I'm pleased to inform you that your manuscript has been deemed suitable for publication in PLOS ONE. Congratulations! Your manuscript is now with our production department. 

Kind regards, 

on behalf of

Dr. Myeongwoo Lee 

Academic Editor

PLOS ONE